# Disaster Risk Reduction Funding: Investment Cycle for Flood Protection in Japan

**DOI:** 10.3390/ijerph19063346

**Published:** 2022-03-11

**Authors:** Mikio Ishiwatari, Daisuke Sasaki

**Affiliations:** 1Graduate School of Frontier Sciences, The University of Tokyo, Kashiwa 277-8563, Japan; 2International Research Institute of Disaster Science, Tohoku University, Sendai 980-8577, Japan; dsasaki@irides.tohoku.ac.jp

**Keywords:** financing mechanism, flood protection, investment cycle, investment in DRR, Japan, long-term plan, lost decades, Sendai Framework for Disaster Risk Reduction

## Abstract

Background: Investment in disaster risk reduction is crucial in order to mitigate disaster damage. However, for many countries, particularly developing ones, financing investment in disaster risk reduction is challenging. This study aims to examine the factors that affect investments in flood protection and the approaches to securing investments by analyzing investment trends in Japan. Methods: This study examines 150 years of flood protection and investment cycles that helped reduce damages in Japan. The dataset of flood protection budgets, flood damage, and national income since 1878 was created from public statistics. Documents and reports concerned with disaster management, river management, and finance were examined. Results: The study found five investment cycles of flood protection from the late 19th century to the present. The country established financing mechanisms, such as legislation and long-term plans, following major flood disasters. However, external shocks such as war, economic recession, disaster, and tightened national finance had a major impact on these investments. The fluctuations in the budget created an investment cycle. The country had increased its budget to 0.9% of its national income in the 1990s. It often experienced flood damage accounting for over 1% of the national income until 1961, but succeeded in decreasing the damage to less than 1%, and currently it is limited to less than 0.4%. Conclusions: The financial mechanisms established from the long-term perspective could support an increase in budgets for flood protection, leading to a decrease in damage. However, established financing mechanisms may weaken the financial flexibility of the country.

## 1. Introduction

Investment in disaster risk reduction (DRR) is crucial for mitigating disaster damage, which is increasing in most parts of the world due to socio-economic and climatic changes [1,2]. Since disasters hinder growth and sustainable development, reducing disaster risks could promote the achievement of the Sustainable Development Goals (SDGs) [3]. The Japanese government identified DRR as one of the priority areas for the promotion of SDGs, in particular making cities resilient, relating to SDG 11, and adapting to disaster risks increased by climate change, relating to SDG 13 [4]. The Sendai Framework for DRR (SFDRR), which UN member states agreed to in 2015, shows the paradigm of DRR and emphasizes investing in DRR as one of the four priority actions [5]. However, continuously increasing investments in DRR is a challenge [6,7,8]. Governments are often forced to reduce flood protection investments or divert parts of this to other sectors because of external factors such as economic recessions and wars. This study examines investment trends and factors that affect investments in flood protection in Japan. It also aims to identify various approaches to securing investment for DRR. 

It is well known that capital investment creates an economic cycle with a period ranging from 7–11 years. This cycle, first identified by the French economist Clement Juglar and called the business cycle, is affected by the timing of capital investment corresponding to the durability of the company’s equipment [9]. Investment in DRR can also create a cycle affected by fluctuating budgets for flood protection and changing flood damage.

Ishiwatari and Sasaki [10], were able to define an investment cycle for flood protection by examining investment trends in Asia’s major flood-prone economies. These economies increased government budgets for flood protection following major flood disasters, but were then unable to sustain these budgets. The share-to-gross domestic product for the budgets of flood protection and flood damage forms cycles. This study analyzes similar investment cycles in Japan’s almost 150 years of flood protection funding. 

### Evolving Mechanisms of Flood Protection in Japan

Recent studies have examined the trends in flood protection in Japan from a wide range of perspectives, such as technology, institutions, environments, and local communities [11,12,13,14,15,16,17,18]. However, only a limited number of studies have examined investment issues. 

Historically, the Japanese have always fought against flooding. The earliest reference to disaster management of floods can be traced to the 4th century when the emperor instructed the construction of the first river dikes with irrigation facilities along the Yodogawa River in Osaka Prefecture [19]. Local communities were historically responsible for protecting their own houses and agricultural lands since, during the 16th century, only Bakufu, the military government headed by shoguns, and federal lords protected the major towns and castles because of limited technical and financial resources [11]. Takahasi [12] examined the changes in the relationship between local communities and the government since the federal period and found that local communities became less involved in flood protection, as river works required higher levels of engineering in the modern period. 

Following the Meiji revolution in the late 19th century, the country started modernizing its socio-economy using Western technology and introduced flood protection technology from the Netherlands and other Western countries. Takei’s [13] review of the evolving trends of technology and institutions in flood protection, since the Meiji era found that governments promoted flood protection works following large-scale flood disasters, but were unable to implement these continuously due to changing political and financial situations. Takahasi and Uitto [14] traced the evolution of river management policies from the Meiji era and stressed that the country was forced to change its policy to include environmental conservation in river management during high growth in the 1970s and 1980s. Kajiwara [15] examined changes in river administration since the Meiji era from the perspectives of legislation, institution, and technology. He found that the Japanese government focused on technological issues led by engineers, integration of flood protection with water resource development through construction of multipurpose dams, and promoting projects via long-term planning. Nakamura and Oki [16] examined the increasing trends in the safety levels of flood protection programs in major rivers since 1910 and argued that Japan’s, flood risk management has experienced a paradigm shift. where technology has become a priority due to social, economic, and climate (flooding) evolution. Nakamura [17] reviewed the modern history of flood protection by analyzing approaches to determining targeted flood volumes regarding the safety levels of floods. The government, because of its financial and social constraints, initially used recorded maximum floods to decide targeted volumes. In the high growth period since the 1960s, the government applied probability analysis to formulate river basin plans and increased the safety levels of flooding. For example, the government is currently implementing projects to protect the Tokyo and Osaka metropolitan areas against once-in-200-year floods in the major rivers. The government developed methods for examining investment efficiency in the 1960s and formulated manuals for the economic analysis of the projects for flood protection. These manuals include the methods developed for estimating benefits by analyzing asset values in flood-affected areas [18].

## 2. Methods

Ishiwatari and Sasaki [10] conceptualized the investment cycle for flood protection by examining investment trends in major flood-prone economies in Asia (Figure 1a). Usually, governments start to increase budgets for flood protection following severe flood disasters and these investments are usually for managing flood damage. Ideally, governments try to maintain the budget at a specific scale and continue to mitigate the damage caused by floods (Figure 1b). In reality, because of socio-economic or political changes, such as wars and economic recession, governments may reduce this budget over time. Again, when severe floods occur, the limited budget is inadequate for managing the disaster damage, and this leads to the start of a new investment cycle. 

The dataset for flood protection budgets, flood damage, and national income (NI) since 1878 used in this study was generated from public statistics. We use NI because government data on gross domestic product, before 1955, were unavailable. The documents and reports of government organizations concerned with disaster management, river management, and finance were examined to study the financial and socioeconomic situations affecting flood protection investment. 

This study examines annual damage and budgets in the share-of-NI to identify investment cycles. The investment cycle for flood protection is determined as follows:Large disasters that triggered an increase in the budget for flood protection were identified, and periods of increasing damage were determined.The periods of increasing budgets following large disasters were determined.The periods of decreasing budgets and damage before the next large disaster were determined.

## 3. Results and Discussion

### 3.1. Five Investment Cycles for Flood Protection in Japan

In this study, the authors found the following five cycles of investment: 1878–1906 Establishing the modernized mechanism of flood protection.1906–1931 Constructing structural frameworks, such as channels and dikes, for major rivers.1931–1945 National land devastation during the wartime regime.1945–1958 Responding to a series of flood disasters.1958–2014 Implementing flood protection during high growth and recession.

The factors forming these five cycles are summarized in Table 1.

#### 3.1.1. Establishing a Modernized Mechanism for Flood Protection (1878–1906)

Flood damage increased as the country modernized. Japan established mechanisms for implementing national flood protection projects based on the enacted river law in 1896 (Figure 2).

##### Increasing Damage (1878–1896)

Japan modernized and developed its economy by introducing Western technology and systems following the Meiji Revolution in the late 19th century. The country developed its light industry, particularly the spinning and yarn-making industries, and built large-scale factories for this. 

Flood damage began to increase because of economic development and modernization. The assets of factories and urban facilities were accumulated in flood prone areas. Forestry in mountainous and hilly regions had diminished during the revolution, leading to an increase in flood volume in rivers. 

Damage and human losses from flooding more than tripled in the 1890s compared to the 1880s [13]. Flood damage in 1885, 1889 and 1893 reached over 4% of NI. The 1885 flood submerged most of Osaka city and affected 270,000 people. In 1889 and 1893, typhoons damaged Wakayama and Okayama prefectures. Flood and tsunami disasters in 1896 caused damage throughout the country, the annual economic damage reaching 11% of NI. 

While the damage caused by floods increased, the national government’s involvement in implementing flood protection measures was limited. The national government conducted river works mainly for navigation to support economic development, whereas prefecture governments and local communities were responsible for flood protection and conducted low-cost works for limited areas with traditional technology. However, prefecture governments and communities were unable to respond to the increasing flood disasters because of their limited technical and financial capacities [20]. 

##### Increasing Budget (1896–1899)

The government launched national flood protection projects following a series of flood disasters in the 1880s and 1890s, and the budget for flood protection reached 0.9% of NI in 1897. The River Law, enacted in 1896, established a scheme for promoting the national government’s flood protection projects. These measures, which prefecture governments and local communities were unable to implement, covered multiple prefectures, needed modern technology, and incurred enormous costs [21]. The landowners of farmlands requested that the Imperial Diet take measures to mitigate the increasing flood damage. Flood disasters reduced the rental income of landowners from tenant farmers, and the government’s income from property taxes decreased. As major voters, the landowners were able to influence the Diet, since the Diet was established in 1890 under restricted voting rights based on property tax payments [13]. Thereafter, flood protection became the second most important infrastructure policy after railroads [14].

The national government became directly involved in measures taken to prevent floods across ten major rivers [22]. For example, the government promoted the construction of a diversion channel in the Yodogawa River from 1896 to 1910 to protect the downtown area of Osaka from flooding, as well as river improvement works in the Chikugogawa River in Kyusyu in the western Japan region from 1896. This period saw the introduction of Western technology for flood protection. Japanese engineers studying civil engineering in Europe led these projects, while the government invited Dutch engineers as advisors. The concept of flood protection involved collecting as much rainfall as possible in the river and discharging it as quickly as possible through the river channels between the dikes. The river channels were widened and excavated, and continuous dikes were built from the mountains to the sea. These systems prevented floods in urban areas and low-lying farmlands and promoted land use for development activities [23].

##### Decreasing Damage (1899–1901)

The flood damage decreased to 0.4% of NI. The government started national projects in six major rivers before 1900 [24]. 

##### Decreasing Budget (1901–1906)

The government shifted its financing to the Russo-Japanese War in 1904 and 1905 and decreased the budget for other sectors, including flood protection. The budget reduced to less than 0.3% of the NI. The government managed a temporary special account for the military during the war from 1904 to 1906. This account, separated from general revenues and expenditures, was established to manage the expenses necessary for military operations until the end of the war [25].

#### 3.1.2. Constructing a Structural Framework in Major Rivers (1906–1931)

Following major disasters in 1907 and 1910, the government established mechanisms for long-term planning and special accounts to secure multi-year commitment for investment in flood protection. However, the budget for flood protection declined because of the reconstruction after the Great Kanto Earthquake in 1923 and in response to the 1929 Great Depression (Figure 3). 

##### Increasing Damage (1906–1910)

Two typhoons hit Yamanashi and Kyoto Prefectures simultaneously in 1907. The economic damage reached 2% of NI that year. The flood disasters in 1910 left around 2500 people dead or missing in the Eastern Japan Region, including Tokyo. Economic damage reached over 4% of NI.

##### Increasing Budget (1910–1914)

Following the 1907 and 1910 disasters, the government formulated its first long-term flood protection plan in 1911. This long-term plan covered 65 rivers and envisaged completing work in 20 major rivers over 18 years as the first phase and 30 rivers as the second phase [26]. For example, a diversion channel project was started in the Arakawa River in 1911 to protect downtown Tokyo, where urbanization was progressing. This plan required 1.7% of the national budget annually. The government enacted the Law for the Special Account of Flood Protection and created a specific account to manage flood protection, separate from the general national account [27]. This account was financed by the national budget, cost-sharing of local governments, and loans from postal savings. 

The long-term plan was implemented as planned in 1911 and 1912. In 1913, the government reduced its budgets for flood protection because of the consolidation of state finances. 

##### Decreasing Damage (1914–1923)

In 1915, the government abolished the special account for flood protection and suspended borrowing funds from postal savings to improve the nation’s financial situation. Postal savings did not increase during this period. The government could not continue the long-term plan because of the delayed progress of work and the financial conditions affected by inflation after World War I. While the first long-term plan envisaged completing work in all 10 rivers by 1921, in reality the work was completed for only two rivers [26]. The government formulated the second long-term plan in 1921, which covered 73 major rivers [13].

##### Decreasing Budget (1923–1931)

The budget for flood protection declined to 0.3% of NI, since the government had to allocate funds for rehabilitation from the Great Kanto Earthquake in 1923 and the 1929 Great Depression. The earthquake killed over 100,000 people in Tokyo and its neighboring areas.

Despite this, the government was able to complete continuous high-dike systems for major rivers, because of the flood protection investments made over almost 30 years. These systems formed the structural framework of the major rivers and substantially decreased flood damage by protecting areas previously inundated in alluvial plains. Examples of this are the diversion channel in Tokyo in the Arakawa River and the Okouchi channel in Niigata in the Shinanogawa River, which became functional in 1924 and 1931, respectively.

#### 3.1.3. Wartime Regime (1931–1945)

In the years 1931 to 1945, no investment cycle for flood protection was formed because the national government funds had to be diverted to finance military expenditures. Even though severe floods had occurred in 1934 and 1935, the government decreased the budget for flood protection, reaching its lowest level in history. This period forms an anti-clockwise, reverse investment cycle (Figure 4).

##### Increasing Damage (1931–1935)

In 1934, the Muroto Typhoon hit Osaka and caused a high tide disaster, leaving nearly 3000 dead or missing. This year’s damage reached 3.9% of NI. In 1935, typhoons and heavy rainfall caused flood disasters in Kyoto, Tohoku, and Kyushu. This year’s damage reached 2.1% of NI.

The government increased public expenditure, including the budget for flood protection, in 1932 and 1933 to overcome deflation [28], but did not increase it following the 1934 flood. The flood protection budgets exceeded 0.5% of NI in 1933 and 1934 and then decreased to approximately 0.3% until the 1940s.

The government could not start projects in 41 out of the 73 rivers targeted by the second long-term plan of 1921 as the priorities had to change because of flood disasters [13]. The government revised the existing long-term plan and formulated a third long-term plan in 1933. This plan envisaged completing projects in 24 rivers in need of urgent repair for 15 years, and included subsidies for the repair of small- and medium-sized rivers and for erosion control managed by prefectures [26]. 

##### Decreasing Budget (1935–1945)

The government could not sufficiently start working on its third long-term plan, because Japan had shifted to a wartime regime with the outbreak of the Second Sino-Japanese War in 1937. The government established and managed a temporary military special account until 1946. The war was long, and the government had to drastically cut general expenditures, including the flood protection budget, and increase war expenditures [25]. The budget for flood protection decreased to 0.2% of NI in 1943 and 1944, the lowest since the Meiji era.

#### 3.1.4. A Series of Flood Disasters (1945–1958)

Japan had to recover from World War II and respond to a series of severe floods that occurred in the 1940s and the 1950s (Figure 5).

##### Increasing Damage (1945–1947)

In 1947, the Kathleen typhoon devastated the Kanto and Tohoku regions. The dikes of the Tonegawa River were broken, and a wide area of the Greater Tokyo was submerged. Some 1000 people died, and this year’s damage reached over 10% of NI.

From 1946 to 1954, Japan’s economic damage exceeded over 2% of NI each year. The average number of dead and missing individuals during this period was over 1300 per year. This damage was unusual, considering that in the 1910s, 1920s and 1930s, that is, before World War II, there were only two years, 1934 and 1935, when the damage exceeded 2%.

Several factors have been reported to have caused such large damage. The government could not allocate sufficient funds for flood protection during the wartime regime. The budget was less than 0.3% of the NI in 1935. Second, the large-scale destruction of mountains removed a natural barrier against floods. During this period, the amount of timber logged almost doubled, mostly consumed by the military (from the 1930s) and for reconstruction following the war. Since major cities were damaged by the war, and food and goods were in short supply, large amounts of lumber were needed for reconstruction [29]. Takahasi [12] pointed out that flood protection measures, since the Meiji era, caused higher peak flood flows in the middle and lower reaches of rivers, increasing flood damage. Continuous high dikes had been constructed to allow floodwaters to drain quickly, but these also retained floodwaters, which previously would overflow upstream into the river channels.

##### Increasing Budget (1947–1953)

The government formulated a national budget focusing on flood protection in 1950 [25]. The budget for flood protection reached 0.68% of NI in 1950.

While the budget increased, the damage did not decrease substantially and remained over 2% of NI. In 1953, three major flood disasters caused by typhoons and heavy rainfall damaged the western Japan, south Kansai, and Tokai regions. This year’s damage reached over 10% of NI.

##### Decreasing Damage (1953–1955)

The cabinet established the Council of Measures for Forest Protection and Flood Protection in 1953 to promote flood management, considering the severe disasters following World War II [30]. The council consists of ministers of finance, construction, forestry, and academic experts. The council formulated basic guidelines for forest protection and flood protection, with costs accounting for over 40 years of budget. Because of the scale, the concerned ministries could not agree to implement this guideline. From 1954, the damage began to decline and stayed at less than 4% of NI.

##### Decreasing Budget (1955–1958)

The budget for flood protection decreased from 0.59% to 0.48% of NI. The government allocated more financing to areas directly related to developing industrial infrastructure, such as roads and ports. The Japanese economy recovered to pre-war levels, and the country planned to expand it further [31]. Additionally, the government adopted a tighter fiscal policy to curb demand from 1954 to 1958.

#### 3.1.5. Implementing Flood Protection during High Growth and Recession (1958–2014)

The government developed financing mechanisms for long-term planning and special accounting with legislation following the 1959 Isewan Typhoon disaster. The country could increase the budget for flood protection for two decades during high economic growth and kept it at the highest level of approximately 0.8% of the NI with damage equivalent to less than 0.4% of the NI for the next two decades (Figure 6).

##### Increasing Damage (1958–1959)

The Isewan Typhoon left 5098 people dead or missing in 1959, the highest number ever recorded for a typhoon disaster. This year’s damage reached 4.8% of NI. The storm surge caused by the typhoon damaged Nagoya City, the Aichi Prefecture, and the Mie Prefecture. The bay area around Nagoya City has been created by land reclamation since the 16th century, and most of the land is located in low-lying lands below sea level. This area was urbanized because of the economic boom after World War I, the military boom during World War II, and economic recovery and expansion after the Korean War. However, disaster-prevention measures for this region had not yet been developed.

##### Increasing Budget (1959–1982)

The government established mechanisms for securing stable budgets to implement flood protection with a long-term perspective following the Isewan Typhoon disaster. The budget for flood protection increased from 0.5% of NI in 1959 to over 0.8% in 1978, and remained at this level until 1982. Areas protected from floods increased to 24% in 1970 and 32% in 1980 [32]. Flood damage decreased to less than 1% of NI during this period, except in 1961. The Second Muroto Typhoon and Baiu rain front damaged 2% of the NI in that year.

The Emergency Measures for Forest Protection and Flood Protection and Special Account for Flood Protection were enacted in 1960. The Cabinet decided on a 10-year plan for flood protection with committed budget plans, which was the first long-term plan following World War II. Before the first plan, draft plans were rejected three times in the 1950s because the Ministry of Finance did not agree on them [18]. The plan aims to complete river works in 100 major rivers throughout the country within 15 years. The budget planned for 10 years accounted for 7% of NI. This amount was decided by shrinking the total estimated project costs without considering national economic growth or the share of the total government budget [33]. In 1960, the first year of the long-term plan, the budget increased by over 40%. The government created a special account to manage budgets for flood protection, which was separate from the general account (Figure 7). This special account received funds from electric companies and the local governments shared some of the costs. In addition, the general account of the national government along with the special account provided budgets for national projects and subsidies to local government.

Since 1959, the Japanese government has adopted an expansionary fiscal policy to promote economic growth and create the foundation for growth. The government rapidly expanded its fiscal scale and increased its fiscal activities, including spending on flood protection. In 1965, the government began issuing deficit covering government bonds to stimulate the economy [25].

The administration system for managing rivers was changed by revising the river law in 1964. The national government is responsible for managing major rivers from the perspective of the river basin, and can improve flood protection in a balanced manner between left and right banks, as well as upstream and downstream. Under the original law, prefectural governors were responsible for managing rivers and could implement protection work, which sometimes caused conflicts with other prefectures in the same basin [34].

Japan needed to respond to drastic socio-economic changes, such as urbanization and development activities, in the 1960s, when the average annual real growth rate of the gross domestic product was over 10%. The long-term plans for flood protection were revised before the end of the planning period to respond to financial and socio-economic changes. Since the national budget expanded and the need for investing in urban areas increased, the government invested 18% more than the planned expenditure on flood protection for the five years between 1960 and 1965 [35]. In 1965, the Cabinet decided on a second long-term plan for five years by revising the first 10-year plan at midyear. The second plan aimed to develop flood protection facilities in a balanced manner nationwide to conserve and develop national lands in response to socio-economic development, stabilize people’s lives, and strengthen the industrial foundation. The Cabinet revised the second plan and decided on a third plan, which increased the planned budget by 85% in 1968, the midyear of the second plan. The Cabinet decided on a fourth plan for five years in 1972, in which the budget was double that of the third plan [35].

The purpose of these long-term plans has evolved. Originally, their aim was to secure budgets, but this later changed to setting long-term goals for flood protection projects [36]. For example, the fourth plan in 1972 included decreasing flood damage in urban rivers and improving the river environment, and the eighth plan in 1991 included disaster management and supporting the local community’s development (Figure 8).

According to these long-term plans, Japan has revised its strategy for flood protection to respond to socio-economical changes. The Japanese government started new programs, increasing investment in urban areas, to mitigate the damage accelerated by urbanization in the 1970s. These urban programs cover software measures, such as evacuation and early warning, and land use regulation, in addition to conversional structural measures [37]. The government began its nature-oriented river management program in 1990 to respond to growing public environmental awareness. Further, the government expanded this program to include green infrastructure programs in response to the Great East Japan Earthquake and Tsunami in 2011 [38]. To adapt to the adverse effects of climate change, the government initiated the new strategy “River Basin Disaster Resilience and Sustainability by All” in 2021, which engages the whole society, including the central government, local government, private sector, civil societies, and local communities. This strategy comprehensively covers various measures: (a) conventional structure of flood protection; (b) exposure reduction by relocation from risk areas; and (c) software measures for warning, evacuation, response, and recovery [18,39].

The government has also started to incorporate science-based planning for flood protection projects. It adapted a probability approach to decide safety levels in the 1950s and applied the flood scale of once-in 80–100-years as the targeted safety levels for major rivers [17]. The Ministry of Construction published technical criteria for designing flood scales using probability analysis in 1958. Before World War II, the government decided on safety levels according to recorded flood volumes.

The safety levels of flood scales designed for protection have increased as the flood protection budget has expanded. The flood scale in major rivers, throughout the country, increased substantially after 1975 [16]. For example, the government is currently promoting works for securing the safety levels of major rivers in metropolitan areas at the safety levels of once in 200- or 150-year scale of floods, following several revisions of design flood volumes. The design flood volume almost doubled in the Yodogawa River, flowing through the Osaka metropolitan area [35].

The development of technology for managing dams has also enabled an increase in safety levels. The Law of Specified Multipurpose Dams was enacted in 1957 to coordinate multiple ministries and water user organizations for multipurpose dam construction [40]. The purpose of multipurpose dams is to supply urban water and protect against flooding. This law aims to promote multi-purpose dam construction by designating the construction minister to implement and manage dams and by clarifying the cost-sharing mechanisms among water users.

##### Decreasing Damage (1982–2000)

The flood damage decreased to less than 0.4% of the NI during this period. The protected area increased from 32% in 1980 to over 50% in 2000.

As the economy continued to stagnate in the early 1990s due to the collapse of the bubble economy, the government began to stimulate the economy through fiscal policy. In 1995, the budget for flood protection reached 0.93% of the NI, the highest on record. 

##### Decreasing Budget (2000–Present)

The government has decreased the budget for flood protection since 2000 because of its severe financial situation. The government applied financial policies to stimulate the economy in the 1990s, but changed to tight fiscal expenditure in 2000. Since the formation of the Koizumi cabinet in 2001, the government has promoted austerity policies focusing on the reduction of public works [41]. The share of the NI in the flood protection budget halved from over 0.8% in 2000 to 0.4% in 2010. Damage was limited to less than 0.2%, except for 0.5% in 2004. In 2003, the Cabinet formulated the Priority Plan for Social Infrastructure Development by integrating nine long-term sectoral plans, including flood protection [42]. The cabinet recognized that the long-term plans for each sector lack flexibility in allocating budgets among sectors and in responding to economic and financial situations [43]. Since the new infrastructure development plans do not include targeted budgets, the government has lost sight of its long-term commitments.

### 3.2. Mechanisms for Securing Budgets for Flood Protection

Factors affecting flood damage and investment for flood protection can be summarized in Figure 9. Major disasters triggered the establishment of financing mechanisms for flood protection. The River Law was enacted in 1896 following major disasters in the 1880s and the 1890s. The law enabled the national government to directly implement flood protection works that local communities and the prefectural government were responsible for initially. The 1910 flood drove the formulation of the first long-term plan for flood protection that mentions multiyear budget targets. Following the Isewan typhoon disaster in 1959, the cabinet began formulating a long-term plan in 1960.

Legislation can provide a foundation for increasing investment in flood protection. Until the enactment of the River Law in 1896, the government could not increase budgets for flood protection, even when severe floods damaged more than 4% of the NI. After enacting this law, the government could increase budgets to develop infrastructure for flood protection following large-scale disasters that caused damage equivalent to more than 4% of the NI with the exception of the 1934 flood during the military expansion period.

The robust mechanism of financing investment enabled stable budgets from 1958 until 2000, even in times of economic recession and other shocks. This is why the last cycle shows longer periods than the other four cycles. The government has secured multi-year financing commitments by developing long-term plans and a special account for flood protection. As the cabinet, including the finance minister, decide on long-term plans involving budget amounts planned for multiple years, annual fluctuations in budgets could be avoided. Long-term plans continued to exist, being formulated nine times from 1960 to 2003 even in periods of worsening national economic and financial situations. The focus of these plans changed from the urgent need to decrease flood damage to environmental improvement and support for the local economy. Thus, Japan succeeded in increasing and maintaining its budget. The budget increased from 0.5% of NI in the late 1950s to approximately 0.9% of NI in the 1990s. Damage was limited to less than 1% of NI in 1962 and further decreased to less than 0.2% of NI in the 2000s, except for 0.5% in 2004. Flood damage decreased continuously during this period.

Securing commitment to a sector means losing flexibility in budget arrangements from the perspective of national finance. The administration led by Prime Minister Koizumi recognized that these mechanisms disturb the restructuring of the national budget and abolished the long-term plans and special accounts for specific sectors, including flood protection, in 2003 [42]. After a high-growth period, Japan has faced severe financial constraints because of economic stagnation since the 1990s.

While legislation and financial schemes can contribute to securing budgets for flood protection, major external shocks have affected the financing of investments. Before World War II, the country established a mechanism for long-term planning and special accounts. However, because of the shocks of wars, large-scale disasters, economic recession, and inflation, the government could not finance flood protection investments as planned. Also, the government is currently reducing flood protection budgets, since the Japanese economy is experiencing a long period of stagnation known as the “lost decades” following the collapse of the bubble economy in 1991 [44].

### 3.3. Implications of Fulfilling SFDRR Targets for Japan

While Japan achieved Target E of SFDRR, formulating DRR strategies at the national and local levels, the country has not reported its baseline and achievements on the other four targets: Target A. mortality; B. people affected; C. economic loss; and D. critical infrastructure [45]. This is because the country has not established definitions and a database. As discussed in this paper, Japan has developed a database for economic damage and mortality caused by flooding for the whole of the previous century. However, databases of damage caused by other natural hazards, such as earthquakes and volcanic eruptions, were not developed properly. Furthermore, there are no clear definitions for people affected and critical infrastructure. We have found a major research gap, which should be filled to develop the database and report its progresses for Target A–D. Furthermore, we consider that national and local government should improve their strategies as Target E of the SFDRR stipulates, by including the issue of financing investment in DRR. Currently, strategies are ignoring this important issue. As this paper discusses, securing budgets for the long-term perspective is crucial for mitigating disaster damage, leading to promotion of SDGs. It is our hope that this study can contribute to finding a clue for further improvement of Japanese DRR strategies in the future.

## 4. Conclusions

This study examines 150 years of flood protection and investment cycles that helped reduce damages in Japan. It examines the factors that can influence investments in flood protection and approaches that can be taken to secure such investments.

The study identified five investment cycles for flood protection and clarified the definitive triggers of major flood disasters in terms of increasing the budget for flood protection in these cycles. Japan has established mechanisms for securing budgets from a long-term perspective following major flood disasters. These mechanisms include legislation for implementing national work, long-term plans decided by the cabinet, and special accounts specifically managed for flood protection. By increasing investment, Japan could substantially reduce flood damage. However, these investments were also influenced by external shocks such as wars, economic recessions, disasters, and tightened national finance. Through these cycles, the country increased its budget to 0.9% of NI in the 1990s. It often experienced flood damage of over 1% of NI until 1961, but succeeded in decreasing damages to less than 1%, and currently this limit is less than 0.2%.

Developing countries, which are suffering from increasing flood damage because of the socioeconomic and climatic changes, apply these mechanisms to secure investment in flood protection. These mechanisms could support sustaining budgets at a certain level, even though the budgets are affected by various shocks.

The Japanese case also shows that established mechanisms lose flexibility due to the overall financial arrangement of the country. Once a national agency secures a multiyear budget for flood protection, the government cannot immediately reduce it even if the national financial situation worsens.

Future studies should examine investment cycles of DRR in other countries. Some developing countries, such as the People’s Republic of China and the Philippines, are increasing their investments in DRR following recent major disasters. Comparing the factors affecting investments in these countries with those in Japan could provide useful lessons for other countries for establishment of financing mechanisms for DRR.

## Figures and Tables

**Figure 1 ijerph-19-03346-f001:**
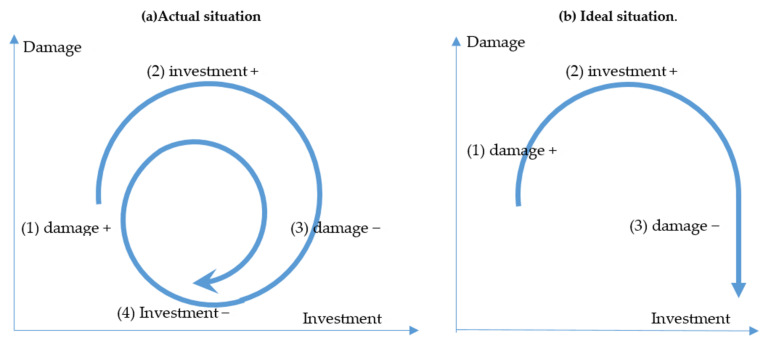
Concept of investment cycle. Source: Authors’ elaboration.

**Figure 2 ijerph-19-03346-f002:**
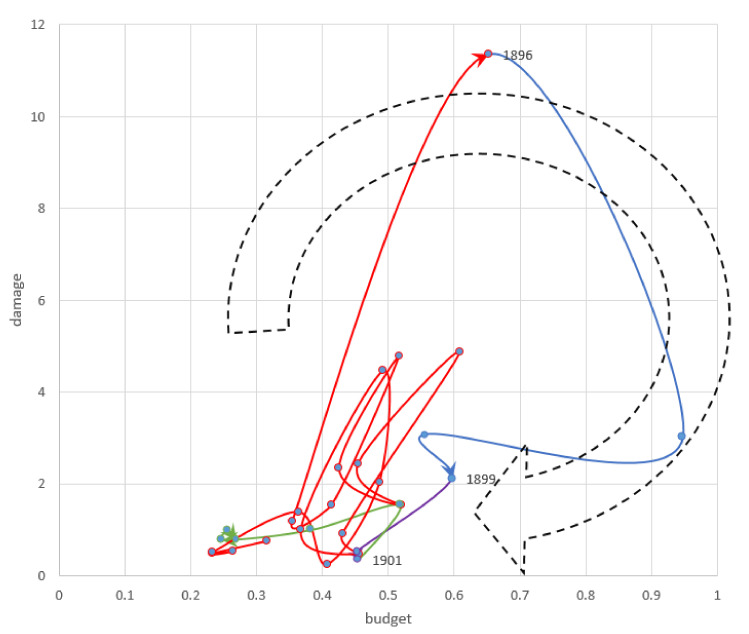
First investment cycle 1878–1906 (Share of the National Income %). Source: Authors’ elaboration.

**Figure 3 ijerph-19-03346-f003:**
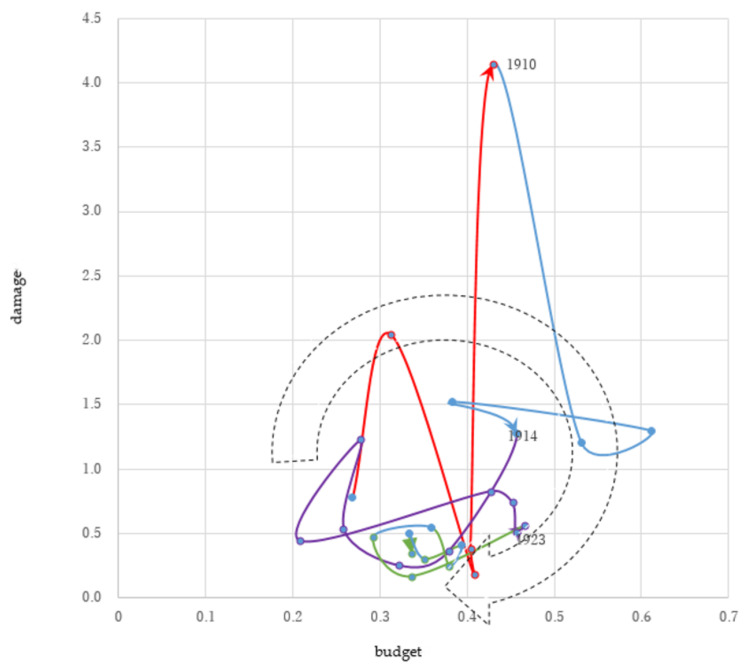
Second investment cycle 1906–1931 (Share of the National Income %). Source: Authors’ elaboration.

**Figure 4 ijerph-19-03346-f004:**
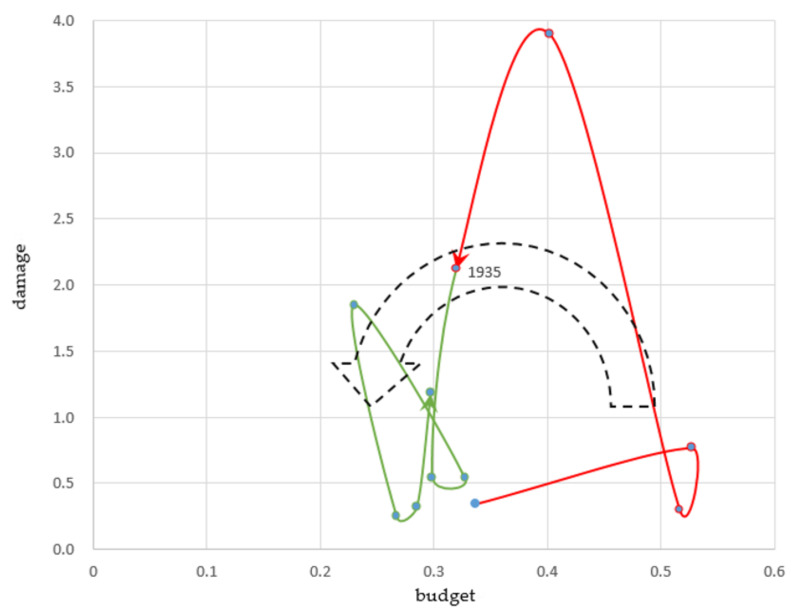
Third investment cycle 1931–1945 (Share of the National Income %). Source: Authors’ elaboration.

**Figure 5 ijerph-19-03346-f005:**
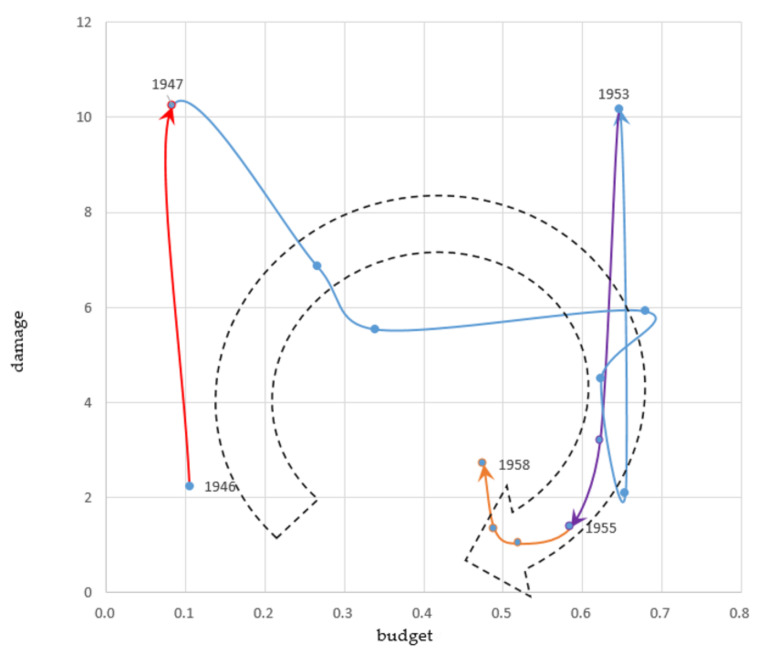
Fourth investment cycle 1945–1958 (Share of the National Income %). Source: Authors’ elaboration.

**Figure 6 ijerph-19-03346-f006:**
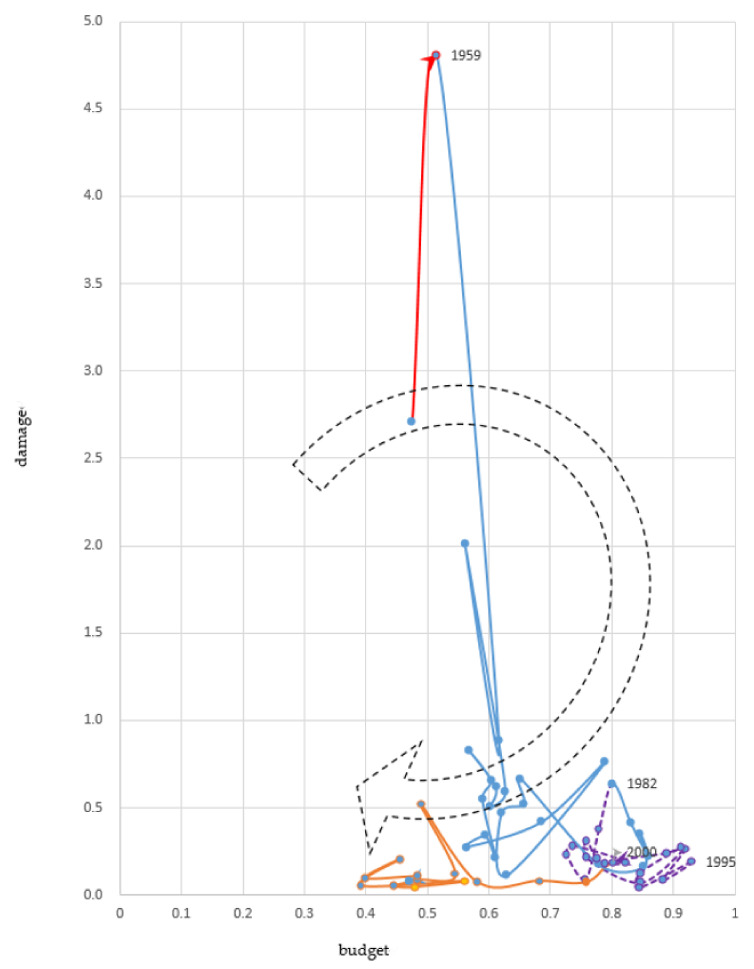
Fifth investment cycle 1958–2014 (Share of the National Income %). Source: Authors’ elaboration.

**Figure 7 ijerph-19-03346-f007:**
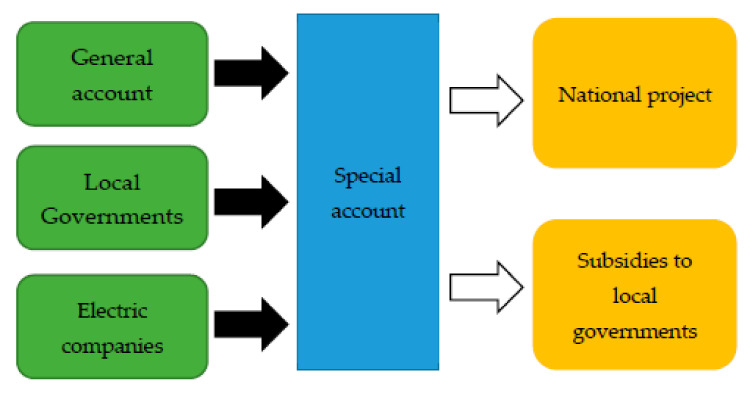
Mechanism of special account for flood protection. Source: Authors’ elaboration.

**Figure 8 ijerph-19-03346-f008:**
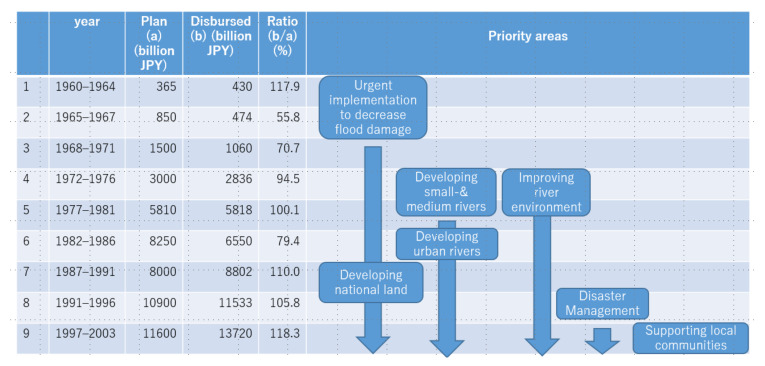
Long-term plans. Source: Modified from Okamura (2002).

**Figure 9 ijerph-19-03346-f009:**
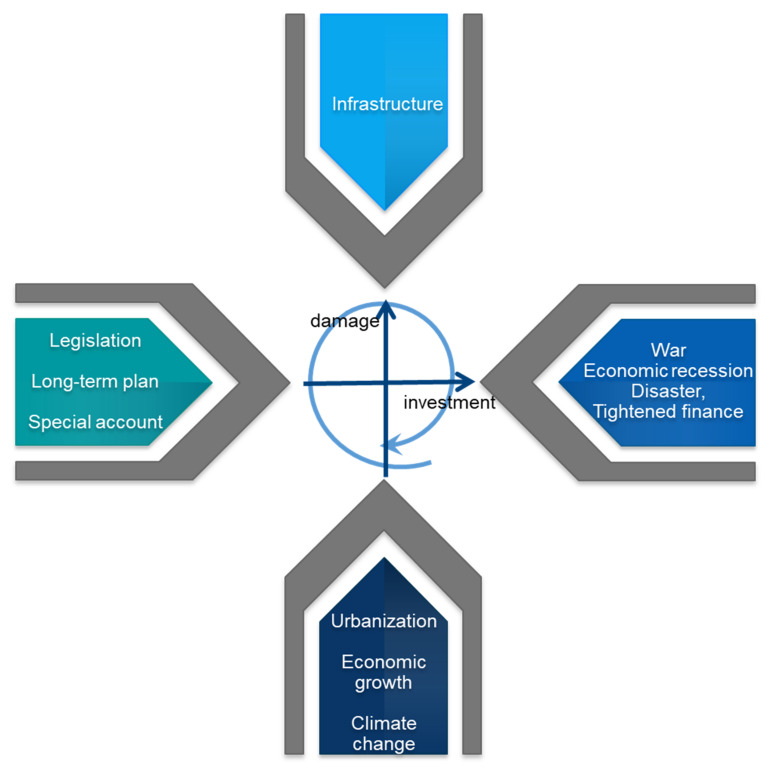
Factors affecting flood damage and investment for flood protection. Source: Authors’ elaboration.

**Table 1 ijerph-19-03346-t001:** Outline of investment cycles.

	(1) Damage Increase	(2) Budget Increase	(3) Damage Decrease	(4) Budget Decrease	Background
	Disasters of Trigger (Share-of-NI)	Instruments to Increase (Share-of-NI)		Causes of Decrease	
I Establishing a modernized mechanism of flood protection					
1878–1906	1878–1896	1896–1899	1899–1901	1901–1906	
	1896 flood (11)	River Law (0.9)		Russo-JPN War	Modernization & Industrialization
II Constructing structural framework in major rivers					
1906–1931	1906–1910	1910–1914	1914–1923	1923–1931	
	1907 & 1910 floods (4)	Long-term plan Special account (0.6)		Kanto Earthquake, Great Depression	Economic growth
III National land devastation in wartime regime					
1931–1945	1931–1935			1935–1945	
	1934 Muroto Typhoon 1935 floods (4)	NA	NA	Sino-JPN War, WWII	Wartime regime
IV Responding to a series of flood disasters					
1945–1958	1945–1947	1947–1953	1953–1955	1955–1958	
	1947, Kathleen typhoon (10)	Responding to major disasters (0.6)		Shift to transport & other sectors	a series of severe floods
V Implementing flood protection during high growth and recession					
1958–2014	1958–1959	1959–1982	1982–2000	2000–2014	
	1959 Isewan Typhoon (5)	Long-term plan Revising river law (0.9)		Economic Recession & Tight national budget	High growth & lost decades

Source: Authors’ elaboration.

## Data Availability

The data that support the findings of this study are available from the corresponding author upon reasonable request.

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
