# Peer review of "Disaster Risk Reduction Funding: Investment Cycle for Flood Protection in Japan"

_ijerph, 2022, doi:10.3390/ijerph19063346_

Round 1

Reviewer 1 Report

This review of financial (capital) investment in Japanese flood protection in itself is useful, but the division into 5 periods seems rather crude, at least from the perspective of the last few decades. Internationally, the approach to flood management has changed dramatically since the late 1980s, and three global disaster conferences have been organised in Japan (Kobe 1995, Hyogo 2005, Sendai 2015), of which Hyogo was strongly shaped by the tsunami just prior. The philosophy of flood protection changed from response to preparation and from dikes to green infrastructure and sustainability. In the meantime Japan has seen many destructive floods and mudslides, most recently in August 2021. Would you claim Japan has not changed its strategy at all in this context? In other words, would you claim that from an investment perspective basically nothing has changed since 1948? It seems unlikely!

I would have expected references to the work of Japanese research organisations ICHARM and RIHN - Have they not published anything of note in this context?

Author Response

This review of financial (capital) investment in Japanese flood protection in itself is useful,

 Thank you for your valuable comments. Your comments have been incorporated comments into the revised draft. Also, professional English editing has been conducted.

but the division into 5 periods seems rather crude, at least from the perspective of the last few decades.

  Thank you for pointing out the important area to be revised.

    • We have highlighted the last cycle by adding the following sentences.

“The robust mechanism of financing investment enabled stable budgets from 1958 until 2000 even in times of economic recessions and other shocks. This is why the last cycle shows longer periods than other four cycles.”

    • To clarify discussions, figure 9 has been added.
    • The paragraphs related have been restructured.

Internationally, the approach to flood management has changed dramatically since the late 1980s, and three global disaster conferences have been organised in Japan (Kobe 1995, Hyogo 2005, Sendai 2015), of which Hyogo was strongly shaped by the tsunami just prior. The philosophy of flood protection changed from response to preparation and from dikes to green infrastructure and sustainability. In the meantime Japan has seen many destructive floods and mudslides, most recently in August 2021. Would you claim Japan has not changed its strategy at all in this context? In other words, would you claim that from an investment perspective basically nothing has changed since 1948? It seems unlikely!

Thank you for suggestion. The following paragraph has been added to explain the changes of strategies for flood protection.

“According to these long-term plans, Japan has revised its strategy of flood protection to respond to socio-economical changes. The Japanese government started new programs increasing investment in urban areas to mitigate the damages accelerated by urbanization in the 1970s. The urban programs cover soft-ware measures, such as evacuation and early warning, and land use regulation in addition to conversional structural measures [37]. The government began the nature-oriented river management program in 1990 to respond to growing public environmental awareness. Further, the government expanded the program to include the green infrastructure programs in response to the Great East Japan Earthquake and Tsunami in 2011 [38]. To adapt to the adverse effects of climate change, the government initiated the new strategy “River Basin Disaster Resilience and Sustainability by All” in 2021, which engages the whole society including the central government, local governments, private sector, civil societies, and local communities. This strategy comprehensively covers various measures: (a) conventional structure of flood protection; (b) exposure reduction by relocation from risk areas; and (c) software measures of warning, evacuation, response, and recovery [18, 39].”

I would have expected references to the work of Japanese research organisations ICHARM and RIHN - Have they not published anything of note in this context?

 Thank you for your advice. The study of Prof. Koike, Director for ICHARM was added as a reference:

Koike T. Evolution of Japan's flood control planning and policy in response to climate change risks and social changes, Water Policy, 2021, 23(S1) 77-84.

“The government developed methods of examining investment efficiency in the 1960s and formulated manuals for economic analysis for the projects of flood protection. The manuals include the methods developed for estimating benefits by analyzing asset values in flood-affected areas.”

“Prior to the first plan, draft plans were rejected three times in the 1950s because the Minis-try of Finance did not agree on them”

“To adapt to effects caused by climate change, the government initiated the new strategy “River Basin Disaster Resilience and Sustainability by All” in 2021, which engages the whole society including the central government, local governments, private sector, civil societies, and local communities. This strategy comprehensively covers various measures: (a) conventional structure of flood prevention; (b) exposure reduction by relocation from risk areas; and (c) software measures of warning, evacuation, response and recovery”

Reviewer 2 Report

Overall this is nice work. Here are my comments and please take them as suggestions for paper improvement:

Please elaborate the author's standing to the conceptual coherence among SDGs, health, and DRR at the beginning of this papert, i.e. either an original one or adopted from other previous works. 

Provide analysis and discussions on Japan Voluntary National Review reports/statements/ key documents submitted regularly to the SDGs mechanism: https://sustainabledevelopment.un.org/memberstates/japan particularly in relation to Goal 3 on health, 11 on sustainable cities, 13 on climate change, and others that may have an actual policy or programmatic linkages. 

Elaborate the relevance of this manuscript and potential implications in informing/accelerating the effort of the Japan government, policymakers, and scientists in fulfilling SFDRR Target E. As can be seen in the SFDRR Monitoring Platform, Japan has not reported its baseline and achievement on Target A -D (https://sendaimonitor.unisdr.org/analytics/country-global-targets/14?countries=85). Hence, this paper can be beneficial by undertaking the following scenarios:

Analyze factors preventing the fulfillment of those targets and potential solution,

Justify the position of national projects already reviewed in the manuscript relative to the SFDRR targets fulfillment. For instance, by scrutinizing the actual policy umbrella that serves as background to those projects and how moving forward those policies can be integrated/streamlined to become a more comprehensive policy on DRR and with relevance to SDGs.

Elaborate more on the author's recommendation on the need to develop a risk management plan at the national plan; particularly how Japan government can accelerate this effort to meet SFDRR deadline of Target E.

If space is available, add discussion on the role, prospect, and current capacity of local government in Japan for creating their own sub-national level DRR policy and strategy. Since this aspect is also covered in SFDRR Target E.

Author Response

Overall this is nice work. Here are my comments and please take them as suggestions for paper improvement:

 Thank you for your valuable comments. These comments have been incorporated into the revised draft. Also, professional English editing has been conducted.

Please elaborate the author's standing to the conceptual coherence among SDGs, health, and DRR at the beginning of this paper, i.e. either an original one or adopted from other previous works.

Thank you for pointing out crucial area of the linkage with SDGs. The following sentence has been added.

“Since disasters hinder growth and sustainable development, reducing disaster risks could promote the achievement of the Sustainable Development Goals (SDGs) [3].”

Provide analysis and discussions on Japan Voluntary National Review reports/statements/ key documents submitted regularly to the SDGs mechanism: https://sustainabledevelopment.un.org/memberstates/japan particularly in relation to Goal 3 on health, 11 on sustainable cities, 13 on climate change, and others that may have an actual policy or programmatic linkages.

The following analysis on the Japanese report has been added.

“The Japanese government identified DRR as one of the priority areas to promote SDGs, in particular making cities resilient on SDG 11 and adapting to disaster risks increased by cli-mate change on SDG 13 [4].”

This reference was added.

“The Government of Japan, Voluntary National Review 2021 Report on the implementation of 2030 Agenda: Toward achieving the SDGs in the post-COVID19 era, 2021. https://sustainabledevelopment.un.org/content/documents/28957210714_VNR_2021_Japan.pdf”

Elaborate the relevance of this manuscript and potential implications in informing/accelerating the effort of the Japan government, policymakers, and scientists in fulfilling SFDRR Target E. As can be seen in the SFDRR Monitoring Platform, Japan has not reported its baseline and achievement on Target A -D (https://sendaimonitor.unisdr.org/analytics/country-global-targets/14?countries=85). Hence, this paper can be beneficial by undertaking the following scenarios:

Analyze factors preventing the fulfillment of those targets and potential solution,

Justify the position of national projects already reviewed in the manuscript relative to the SFDRR targets fulfillment. For instance, by scrutinizing the actual policy umbrella that serves as background to those projects and how moving forward those policies can be integrated/streamlined to become a more comprehensive policy on DRR and with relevance to SDGs.

Elaborate more on the author's recommendation on the need to develop a risk management plan at the national plan; particularly how Japan government can accelerate this effort to meet SFDRR deadline of Target E.

If space is available, add discussion on the role, prospect, and current capacity of local government in Japan for creating their own sub-national level DRR policy and strategy. Since this aspect is also covered in SFDRR Target E.

Thank you again for the important suggestion. The new section “3.3 Implications in fulfilling SFDRR Targets for Japan” has been added. It examined the progress of SFDRR targets, analyses the issues of monitoring, and recommends measures.  

“3.3   Implications in fulfilling SFDRR Targets for Japan

While Japan achieved Target E of SFDRR formulating DRR strategies at the national and local levels, the country has not reported its baseline and achievements on the other four targets: Target A. mortality; B. people affected; C economic loss; and D. critical infra-structure [45]. This is because the country has not established definitions and database. As discussed in this paper, Japan has developed a database for economic damage and mortality caused by flood for over the last the century. However, database of damage caused by other natural hazards, such as earthquakes and volcanic eruptions, were not developed properly. Furthermore, there is no clear definitions for people affected and critical infrastructure. We have found a major research gap, which should be filled to develop the database and report its progresses for Target A–D. Furthermore, we consider that national and local governments should improve their strategies that Target E of the SFDRR stipulates, by including the issue of financing investment in DRR. Currently, the strategies are missing this important issue. As this paper discusses, securing budgets from the long-term perspective is crucial for mitigating disaster damage, leading to pro-motion of SDGs. It is our hope that this study could contribute to finding a clue to further improvement of Japanese DRR strategies in the future.”
